microbiology/molecular biology/genomics

symbiosis, interaction, algae, RNAi, dicer, siRNA

**Authors for correspondence:**
Benjamin H. Jenkins
e-mail: ben.jenkins@zoo.ox.ac.uk
Thomas A. Richards
e-mail: thomas.richards@zoo.ox.ac.uk

†These authors contributed equally.

# Characterization of the RNA-interference pathway as a tool for reverse genetic analysis in the nascent phototrophic endosymbiosis, *Paramecium bursaria*

Benjamin H. Jenkins[1,2,†], Finlay Maguire[3,†],
Guy Leonard[1,2], Joshua D. Eaton[1], Steven West[1],
Benjamin E. Housden[1], David S. Milner[1,2]
and Thomas A. Richards[1,2]

[1]Living Systems Institute and Biosciences, University of Exeter, Devon EX4 4QD, UK
[2]Department of Zoology, University of Oxford, 11a Mansfield Road, Oxford OX1 3SZ, UK
[3]Faculty of Computer Science, Dalhousie University, 6050 University Ave, Halifax, Nova Scotia, Canada B3H 1W5

BHJ, 0000-0002-3803-6418; GL, 0000-0002-4607-2064;
DSM, 0000-0003-3669-7463; TAR, 0000-0002-9692-0973

Endosymbiosis was fundamental for the evolution of eukaryotic complexity. Endosymbiotic interactions can be dissected through forward- and reverse-genetic experiments, such as RNA-interference (RNAi). However, distinguishing small (s)RNA pathways in a eukaryote–eukaryote endosymbiotic interaction is challenging. Here, we investigate the repertoire of RNAi pathway protein-encoding genes in the model nascent endosymbiotic system, *Paramecium bursaria–Chlorella* spp. Using comparative genomics and transcriptomics supported by phylogenetics, we identify essential proteome components of the small interfering (si)RNA, scan (scn)RNA and internal eliminated sequence (ies)RNA pathways. Our analyses reveal that copies of these components have been retained throughout successive whole genome duplication (WGD) events in the *Paramecium* clade. We validate feeding-induced siRNA-based RNAi in *P. bursaria* via knock-down of the splicing factor, *u2af1*, which we show to be crucial to host growth. Finally, using simultaneous knock-down 'paradox' controls to rescue the effect of *u2af1* knock-down, we demonstrate that feeding-induced RNAi in *P. bursaria* is dependent upon a core pathway of

host-encoded *Dcr1*, *Piwi* and *Pds1* components. Our experiments confirm the presence of a functional, host-derived RNAi pathway in *P. bursaria* that generates 23-nt siRNA, validating the use of the *P. bursaria*–*Chlorella* spp. system to investigate the genetic basis of a nascent endosymbiosis.

## 1. Introduction

Endosymbiosis was fundamental for the evolution of eukaryotic cellular complexity [1–4]. In order to investigate the genetic basis of an emergent endosymbiotic system, we must develop experimentally tractable endosymbiotic model species [5–7]. *Paramecium bursaria* is a ciliate protist which harbours several hundred cells of the green algae, *Chlorella* spp., in a nascent and facultative photo-endosymbiosis [8–12]. The algae provide sugar and oxygen derived from photosynthesis, in exchange for amino acids, $CO_2$, divalent cations and protection from viruses and other predators [5,6,13–19]. While the interaction is heritable, the *P. bursaria*–*Chlorella* spp. system is described as a 'nascent' or 'facultative' endosymbiosis, as both host and endosymbiont can typically survive independently [10,20–23]. *Paramecium bursaria* therefore represents a potentially tractable model system with which to investigate the genetic basis of a nascent endosymbiotic cell–cell interaction.

RNA-interference (RNAi) is a form of post-transcriptional gene silencing that is dependent upon conserved small (s)RNA processing pathways [24–26]. The principal components of a functional RNAi pathway are conserved in many eukaryotes [27–29], though loss in some lineages suggests a mosaic pattern of pathway retention [30]. Typically, these pathways rely on size-specific sRNA processing via an endoribonuclease Dicer [26,27], targeted RNA cleavage activity of an Argonaute (AGO-Piwi) containing effector complex [28,31] and RNA-dependent RNA polymerase (RdRP) amplification of either primary or secondary sRNA triggers [32–37]. These triggers may include partially degraded mRNA cleavage products [36], exogenous sRNA [36], or full-length mRNA transcripts [32], suggesting that RdRPs may have broader functions in some systems.

In ciliates, multiple whole genome duplication (WGD) events have led to the rapid expansion of gene families encoding RNAi components [38–40], resulting in a subsequent diversification of protein function. Example functions include transposon elimination, nuclear rearrangement and transcriptional regulation [32,35,41–45]. Elegant investigation of the non-photo-endosymbiotic model system, *Paramecium tetraurelia*, has identified three distinct classes of RNAi pathway in *Paramecium*. The ciliate-specific scan (scn)RNA (25-nt) and internal eliminated sequence (ies)RNA (approx. 28-nt) pathways are endogenous, and function primarily to eliminate the bulk of non-coding DNA present in the germline micronuclear genome during development of the somatic macronucleus [35,41–43,46,47]. *Paramecium* also encodes a short-interfering (si)RNA (23-nt) pathway capable of processing both exogenously [48–51] and endogenously [32,44] derived RNA precursors. Although siRNA is believed to have evolved to protect against foreign genetic elements (such as viruses, transposons and transgenes) [52], some siRNA-based RNAi factors have also been implicated in the regulation of endogenous transcriptome expression in the non-photo-endosymbiotic model system, *P. tetraurelia* [32].

In the photo-endosymbiotic *P. bursaria*–*Chlorella* spp. system, the existence of a functional siRNA-based RNAi pathway would provide an experimental approach to knock-down gene expression via the delivery of exogenously derived double-stranded (ds)RNA homologous to a target transcript [33,48]. Preliminary evidence suggests that siRNA-based RNAi can be induced in *P. bursaria* (strain 110 224) [53]; however, a comprehensive analysis with appropriate controls has yet to be conducted. To demonstrate direct evidence of an RNAi-mediated effect, one would need to rescue a putative phenotype through targeted inhibition of the RNAi knock-down machinery. Such controls are of paramount importance when conducting genetic knock-down experiments in a complex endosymbiotic system, and the presence of a eukaryotic green algal endosymbiont in *P. bursaria* necessitates caution. RNAi has been reported in some green-algal species [54], and thus it is important that controlled experimental characterization of these distinct pathways be conducted before genetic knock-down in *P. bursaria* can be inferred.

Here, we elucidate a cognate repertoire of predicted RNAi component-encoding genes present in *P. bursaria*, confirming that the host genome encodes essential proteome constituents of the siRNA-, scnRNA- and iesRNA-based RNAi pathways. These include multiple paralogues of the pathway components; Dicer, Dicer-like, Piwi (AGO-Piwi), Rdr (RdRP), Cid and Pds1, which have been identified in the non-photo-endosymbiotic model system, *P. tetraurelia* [33,34,36]. We trace the occurrence of RNAi protein-encoding genes in the *Paramecium* clade using comparative genomics combined with transcriptomics and further resolved by phylogenetic analysis, and demonstrate that

these genes have been retained throughout successive WGDs. Using an *E. coli* vector feeding-based approach for RNAi induction, we demonstrate functional siRNA-based RNAi in *P. bursaria* via knock-down of a conserved ciliate splicing factor, *u2af1*, which we show to be similar to the *u2af* (65 kDa) constitutive splicing factor present in humans [55,56]. We demonstrate that RNAi-mediated knock-down of *u2af1* results in significant culture growth retardation in *P. bursaria*, suggesting that this gene has a critical function. Finally, we corroborate the function of several siRNA-based RNAi factors in *P. bursaria*; including *Dcr1*, two unduplicated AGO-Piwi factors (*PiwiA1* and *PiwiC1*) and a *Paramecium*-specific *Pds1*, via simultaneous component knock-down to rescue *u2af1* culture growth retardation. Collectively, these data support the presence of a functional, host-derived, exogenously induced siRNA-based RNAi pathway in the *P. bursaria*–*Chlorella* spp. endosymbiotic system, dependent on *Dcr1*, *Piwi* and *Pds1* protein function.

# 2. Results

## 2.1. Bioinformatic identification of a putative RNAi pathway in *P. bursaria*

A feeding-induced siRNA-based RNAi pathway has been validated as a tool for gene knock-down in the non-photo-endosymbiotic ciliate, *P. tetraurelia* [33,34,36,48]. To establish the presence of a comparable pathway in *P. bursaria*, combined genomic and transcriptomic analyses were employed to identify putative homologues for all previously characterized *P. tetraurelia* RNAi protein components [33,45] (figure 1). We found that *P. bursaria* encodes a total of five Dicer or Dicer-like endonucleases (*Dcr1*, *Dcr2/3*—electronic supplementary material, dataset S1; *Dcl1/2*, *Dcl3/4* and *Dcl5*—electronic supplementary material, dataset S2), three RdRPs (*Rdr1/4*, *Rdr2* and *Rdr3*—electronic supplementary material, dataset S3), six AGO-Piwi components (*PiwiA1*, *PiwiA2*, *PiwiB*, *PiwiC1*, *PiwiC2* and *PiwiD*—electronic supplementary material, dataset S4), a single *Paramecium*-specific Pds1 (*Pds1*—electronic supplementary material, dataset S5), and two nucleotidyl transferase (*Cid1/3* and *Cid2*—electronic supplementary material, dataset S6) genes. Among those identified are homologues of the essential feeding-induced siRNA pathway components present in *P. tetraurelia*. In *P. bursaria*, these are; *Dcr1*, *Pds1*, *Rdr1/4*, *Rdr2*, *Cid1/3*, *Cid2*, and putative *PiwiA1* and *PiwiC1* homologues, although we were unable to accurately identify the precise relationship of the *Piwi* paralogues due to lack of phylogenetic resolution (electronic supplementary material, dataset S4). Sequences corresponding to each of these RNAi protein-encoding genes were present in our *P. bursaria* transcriptome dataset, indicating that these host-derived RNAi components are transcriptionally active. These data reveal that *P. bursaria* encodes a putative functional feeding-induced siRNA pathway, indicating that an experimental approach to knock-down gene expression is tractable in this system. Additionally, we show that *P. bursaria* encodes homologues for components of the transgene-induced siRNA pathway, as well as the endogenous ciliate-specific scnRNA and iesRNA pathways involved in nuclear reorganization and development. For a full list of identified RNAi components, and predicted associated pathways in *P. bursaria*, see table 1.

Further analyses were conducted to identify the presence of comparable RNAi pathway components in the algal endosymbiont. For a full overview of the host and endosymbiont transcriptome dataset binning process, please refer to the Methods section. Using both *P. tetraurelia* and *C. reinhardtii* query sequences, our analyses identified a putative homologue for *Dcl1* that clustered with strong support to known green algal Dicer sequences (electronic supplementary material, dataset S7). No algal homologue for AGO-Piwi or RdRP could be detected in any of the algal-endosymbiont datasets sampled, suggesting that these components are either not transcriptionally active, or are absent altogether in the algal endosymbionts of *P. bursaria* sampled here. The absence of RdRP is consistent with its absence in most green algal species sampled [54]. To explore the possible function of a *Dcl* homologue in the sampled endosymbiotic green algae of *P. bursaria*, we conducted sRNA sequencing of algae isolated from the host under standard growth conditions. Our sRNA sequencing data demonstrated that the isolated algal endosymbiont of *P. bursaria* was not actively generating sRNA greater than 20-nt (electronic supplementary material, figure S1), confirming that the length of endosymbiont Dicer-derived sRNA does not resemble those of siRNA (23-nt), scnRNA (25-nt) or iesRNA (approx. 28-nt) known to be generated by the non-photo-endosymbiotic model system, *P. tetraurelia* [33–35,41,42]. This is an important distinction, as it would allow one to ensure that any genetic knock-down approach in the *P. bursaria*–*Chlorella* spp. system could be attributed to the *Paramecium* host, based on the size of sRNAs generated.

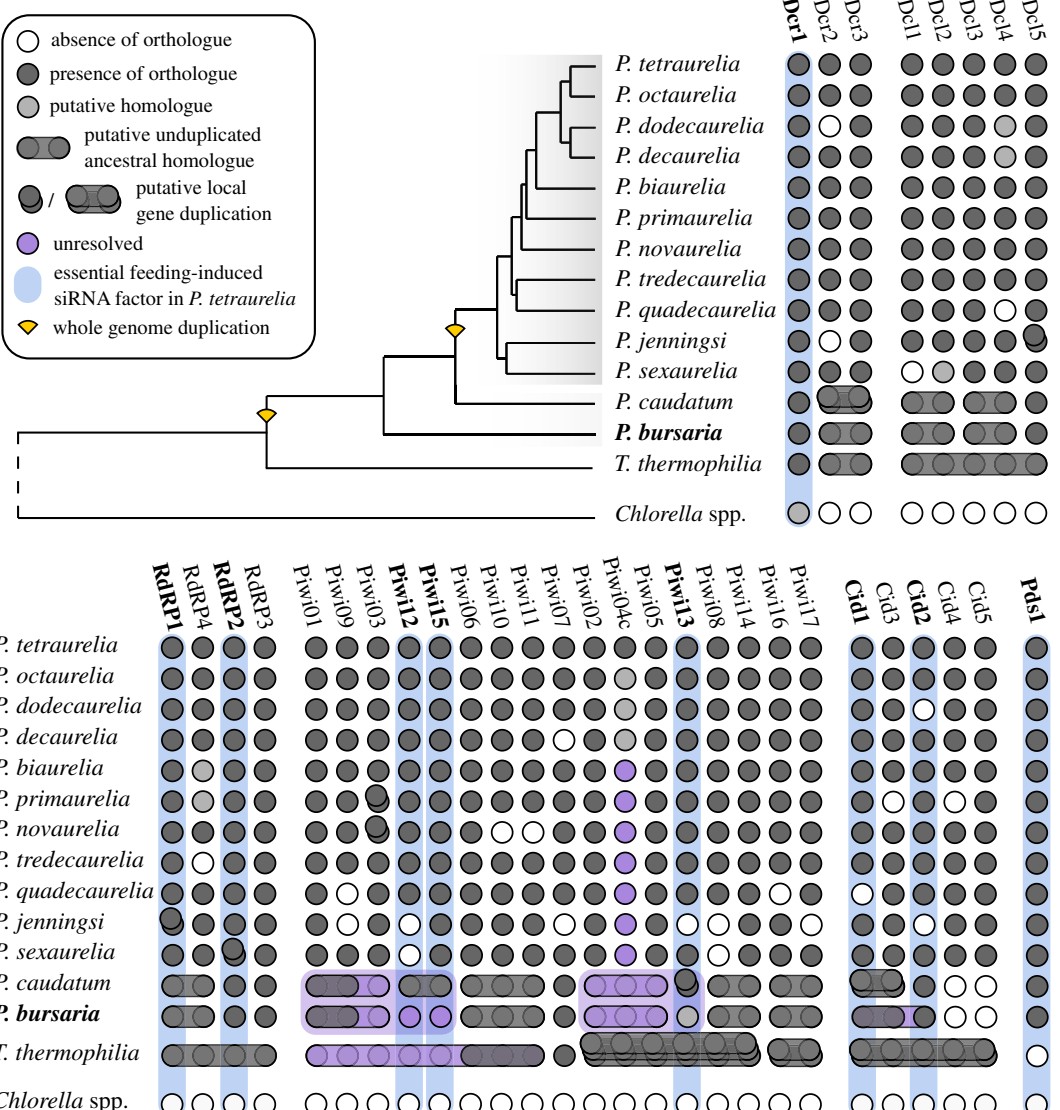

**Figure 1.** Identifying a putative RNAi pathway in *P. bursaria*. Coulson plot showing the presence/absence of putative RNAi pathway component-encoding genes, identified from *Paramecium* genome/transcriptome sequence surveys based on shared sequence identity. Genes highlighted in blue represent components essential for feeding-induced siRNA-based RNAi in *P. tetraurelia*. Genes highlighted in purple represent duplicated components with unclear paralogue/orthologue resolution. Horizontally 'merged' genes indicate a putative unduplicated ancestral homologue. Vertically 'stacked' genes (single or unduplicated orthologues) represent putative local gene duplications. Phylogeny schematic based on *Dcr1* amino acid alignment (electronic supplementary material, dataset S1), with shaded regions indicating species hypothesized to share the same number of ancestral whole gene duplications (WGDs). For all phylogenies, see electronic supplementary material, datasets S1–6. Nucleotide sequence (https://doi.org/10.6084/m9.figshare.13387811.v1) and amino acid alignment data (https://doi.org/10.6084/m9.figshare. 13387631.v1) for putative *P. bursaria* homologues are available at Figshare.

## 2.2. Validation of feeding-based RNAi in *Paramecium bursaria*

To demonstrate the activity of the putative siRNA-based RNAi pathway in *P. bursaria*, identified in figure 1, we targeted the conserved splicing factor encoding gene, *u2af* [55,56]. Many ciliates genomes are intron-rich and dependent upon splicing for transcription [58,59], thus it was predicted that knock-down of *u2af* would considerably impact *P. bursaria* growth. Transcriptome analysis revealed that *P. bursaria* encodes three paralogues with sequence similarity to the *u2af* (65 kDa) constitutive splicing factor present in humans [55,56], and indicates that these paralogues probably diverged prior to the radiation of the ciliate clade (figure 2a). Interestingly, the *u2af1* orthologue has been subject to gene duplication prior to diversification of the *Paramecium aurelia* species complex, consistent with a WGD event at the same node, with greater

**Table 1.** Full list of identified RNAi components and predicted associated pathways in *P. bursaria*.

| *Paramecium bursaria* homologue | GenBank accession no.[a] | corresponding *P. tetraurelia* homologue[b] | associated pathway in *P. tetraurelia* | reference |
|---|---|---|---|---|
| Dcr1 | MW715702 | Dcr1 | feeding-induced siRNA (23-nt), transgene-induced siRNA (23-nt) | [33] |
| Dcr2/3 | MW715703 | Dcr2 | — | — |
| | | Dcr3 | — | — |
| Dcl1/2 | MW715704 | Dd1 | — | [35,41,42] |
| | | Dd2 | scnRNA (25-nt) | [35,41,42] |
| Dcl3/4 | MW715705 | Dd3 | scnRNA (25-nt) | [35,41,42] |
| | | Dd4 | — | — |
| Dcl5 | MW715706 | Dcl5 | iesRNA (approx. 28-nt) | [41,42] |
| Pds1 | MW732644 | Pds1 | feeding-induced siRNA (23-nt) | [33] |
| Rdr1/4 | MW715707 | Rdr1 | feeding-induced siRNA (23-nt) | [33] |
| | | Rdr4 | — | — |
| Rdr2 | MW715708 | Rdr2 | feeding-induced siRNA (23-nt), transgene-induced siRNA (23-nt), endogenous siRNA (23-nt) | [32–34] |
| Rdr3 | MW715709 | Rdr3 | transgene-induced siRNA (23-nt), endogenous siRNA (23-nt) | [32–34] |
| PiwiA1 | MW715710 | Ptiwi01 | scnRNA (25-nt) | [57] |
| | | Ptiwi09 | scnRNA (25-nt) | [57] |
| | | Ptiwi03 | — | — |
| | | Ptiwi12 | feeding-induced siRNA (23-nt) | [33] |
| | | Ptiwi15 | feeding-induced siRNA (23-nt) | [33] |
| PiwiA2 | MW715712 | Ptiwi06 | — | — |
| | | Ptiwi10 | — | — |
| | | Ptiwi11 | — | — |

**Table 1.** (Continued.)

| Paramecium bursaria homologue | GenBank accession no.[a] | corresponding P. tetraurelia homologue[b] | associated pathway in P. tetraurelia | reference |
|---|---|---|---|---|
| PiwiB | MW715713 | Ptiwi07 | — | — |
| PiwiC1 | MW715714 | Ptiwi02 | — | — |
| | | Ptiwi04c | — | — |
| | | Ptiwi05 | — | — |
| | | Ptiwi13 | feeding-induced siRNA (23-nt), transgene-induced siRNA (23-nt) | [33] |
| PiwiC2 | MW715715 | Ptiwi08 | — | — |
| | | Ptiwi14 | transgene-induced siRNA (23-nt) | [33] |
| PiwiD | MW715716 | Ptiwi16 | — | [33] |
| | | Ptiwi17 | — | [45] |
| Cid1/3 | MW732645 | Cid1 | feeding-induced siRNA (23-nt) | [33] |
| | | Cid3 | transgene-induced siRNA (23-nt) | [33] |
| Cid2 | MW732646 | Cid2 | feeding-induced siRNA (23-nt), transgene-induced siRNA (23-nt) | [33] |

[a]Nudeotide (https://doi.org/10.6084/m9.figshare.13387811.v1) and amino acid (https://doi.org/10.6084/m9.figshare.13387631.v1) sequence data for P. bursaria RNAi component-encoding genes are also available on Figshare.

[b]For phylogenetic identification of Paramecium homologues, see electronic supplementary material, datasets S1–S6.

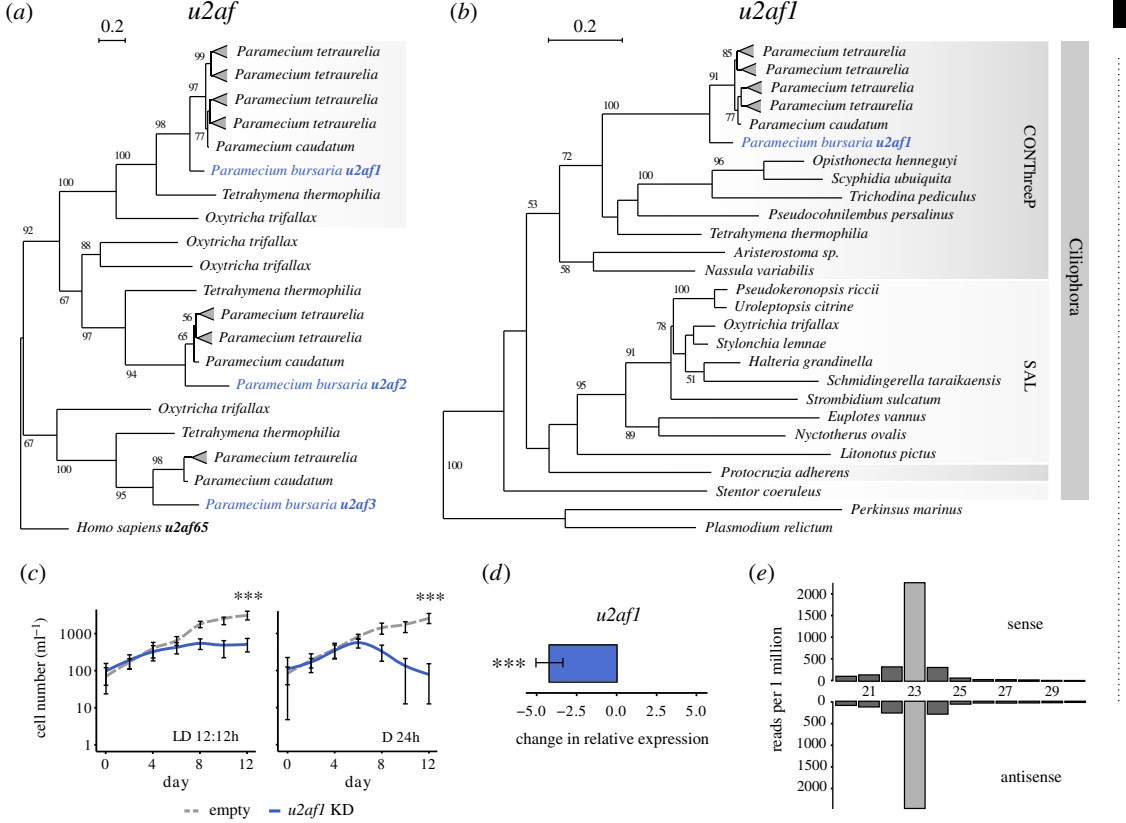

**Figure 2.** Validation of feeding-based RNAi in *P. bursaria*. (*a*) *U2af* phylogeny (based on 166 aligned amino acid sites sampled from the alignment) calculated using IQ-TREE with an rtREV + G4 best fit substitution model chosen according to BIC (Bayesian Inference Criterion implemented in IQ-TREE), and with 1000 non-rapid, non-parametric bootstrap replicates. This phylogeny highlights the three orthologues of *u2af* (65 kDa) encoded by *Paramecium*. Note the shaded branch corresponding to the *u2af1* orthologue targeted in this study. (*b*) Specific *u2af1* phylogeny (based on 225 sampled aligned amino acid sites) calculated using IQ-TREE with an LG + G4 best fit substitution model chosen according to BIC, and with 1000 non-rapid, non-parametric bootstrap replicates. This phylogeny shows the distribution of *u2af1* across the ciliates. Ciliate clades CONThreeP (Colpodea, Oligohymenophorea, Nassophorea, Prostomatea, Plagiopylea and Phyllopharyngea) and SAL (Spirotrichea, Armaphorea and Listomatea) are defined according to Lynn [60,61] and Adl [60,62]. For all phylogenies, bootstrap values above 50 are shown. Amino acid alignment data for putative *P. bursaria* homologues used in the above datasets are available on Figshare (https://doi.org/10.6084/m9.figshare.13387631.v1). (*c*) *Paramecium bursaria* cell number in cultures fed with HT115 *E. coli* expressing *u2af1* dsRNA (blue) or an empty vector control (dashed grey). *Paramecium bursaria* cells were resuspended daily into fresh feeding media for 12 days under standard light–dark (LD 12 : 12 h) or constant darkness (D 24 h) conditions. Note that the effect of *u2af1* dsRNA exposure was more potent when feeding was conducted under constant darkness, giving rise to a mean cell number after 12 days that was 84.4% less compared with parallel cultures maintained under standard light–dark conditions. Data are represented as mean ± s.d. of five biological replicates. Asterisks displayed above the plot denote significant difference in cell number between cultures exposed to *u2af1* dsRNA and an empty vector control at day 12, calculated as ***$p \leq 0.001$ using a generalized linear model with quasi-Poisson distribution. Here and elsewhere, the term 'KD' is used in figure to denote 'knock-down'. (*d*) qPCR of mRNA extracted from day 3 of *u2af1*-RNAi feeding, revealing potent gene knock-down in *P. bursaria* in response to *u2af1* dsRNA exposure. Change in relative expression (ddCT) was calculated for treated (*u2af1* dsRNA) versus untreated (empty vector) control cultures, and normalized against the standardized change in expression of a *GAPDH* housekeeping gene. Data are represented as mean ± s.e.m. of three biological replicates, per treatment. Unpaired dCt values were pooled and averaged prior to calculation of ddCt. Error bars for ddCt values were propagated from the s.e.m. of dCt values. Raw Ct, dCt and error propagation calculations for all ddCt values are available in electronic supplementary material, table S3. Significance for qPCR data calculated as ***$p \leq 0.001$ using a paired *t*-test. (*e*) Size distribution of sense and antisense sRNAs mapping to a 450-nt 'scramble' dsRNA construct, expressed via transformed *E. coli* fed to *P. bursaria* for three days prior to sRNA extraction and sequencing. Scramble dsRNA presented no significant hits to the identified *P. bursaria* host or endosymbiont transcriptome to ensure that the 23-nt sRNA detected was of definitive exogenous origin.

than 85% copy retention across all species sampled ([$n = 39$ retained/44 predicted], showing five putative gene losses in 11 taxa; electronic supplementary material, dataset S8).

Phylogenetic analysis indicates that *u2af1* is highly conserved in ciliates, supporting the hypothesis that it may have an essential function (figure 2*b*). Using an *E. coli* vector-based feeding approach for

RNAi induction, delivery of a 500-nt dsRNA fragment corresponding to *u2af1* resulted in significant *P. bursaria* culture growth retardation compared with an empty vector control, consistent with an RNAi effect (figure 2*c*). Interestingly, retardation to culture growth in response to *u2af1* dsRNA exposure was greater under constant darkness (D 24 h), with a mean cell number after 12 days that was significantly less (−84.4%; ***) compared with parallel cultures maintained under standard light–dark (LD 12 : 12 h) conditions. This is consistent with an increased rate of *P. bursaria* feeding in the dark resulting in greater *E. coli* uptake [63] and therefore increased RNAi potency. Using mRNA extracted from *P. bursaria* during *u2af1*-RNAi feeding, qPCR revealed a significant reduction in *u2af1* gene expression in response to complementary dsRNA exposure (figure 2*d*).

Next, we designed a 450-nt 'scramble' dsRNA control using a 'DNA shuffle' tool (https://www.bioinformatics.org/sms2/shuffle_dna.html) to randomly shuffle a 450-nt nonsense sequence, ensuring that the resultant 'scramble' dsRNA bore no significant sequence similarity to any *P. bursaria* host or algal endosymbiont transcripts present in the transcriptome datasets. For confirmation of the null effect of 'scramble' dsRNA exposure compared with an empty vector control, see electronic supplementary material, figure S2. Following 'scramble' dsRNA exposure, sRNA isolated from *P. bursaria* was sequenced and mapped against the original 'scramble' DNA template. This allowed us to demonstrate a distinct abundance of sense and antisense 23-nt reads in *P. bursaria* (figure 2*e*). These results are consistent with previous studies demonstrating Dicer-dependent cleavage of dsRNA into 23-nt fragments in *C. elegans* and *P. tetraurelia* [26,35]. Collectively, these data confirm the presence of a Dicer-mediated siRNA-based RNAi pathway capable of processing exogenously derived dsRNA into 23-nt siRNA, and which is induced through the consumption of bacterial cells via phagocytosis.

## 2.3. Investigating *Dcr1* function

Having demonstrated feeding-based RNAi induction, we investigated putative Dicer function in *P. bursaria*. Further dsRNA constructs (Dcr1A, Dcr1B) were designed to specifically target two regions of the *Dcr1* transcript present in *P. bursaria*. A BLASTn search against the *P. bursaria*–*Chlorella* spp. host and endosymbiont transcript datasets confirmed that the identified dsRNA template from these constructs was predicted to target only *Dcr1*, accounting for all possible 23-nt fragments and allowing for less than or equal to 2-nt mismatches. Using mRNA extracted from *P. bursaria* during *Dcr1*-RNAi feeding, qPCR revealed knock-down of *Dcr1* in response to Dcr1A and Dcr1B dsRNA exposure (figure 3*a*). Importantly, we found that knock-down was only detected in *Dcr1* when the qPCR amplicon was located directly adjacent to the dsRNA target site, with detectable mRNA reduction less evident as the qPCR target amplicon was moved further along the transcript towards the 5′ end. This finding suggests that the transcript is only partially degraded upon dsRNA-mediated knock-down— an important consideration for the design of effective RNAi reagents for further experiments.

We next checked for the occurrence of any off-target effects arising from Dcr1A/Dcr1B dsRNA exposure, as these may result in knock-down of additional Dicer or Dicer-like components in *P. bursaria*. An additional set of qPCR amplicons was designed to target each of the *Dcr2/3*, *Dcl1/2*, *Dcl3/4* and *Dcl5* transcripts identified from our host transcript dataset (figure 1). Full-length sequences were derived from genomic data and compared with respective transcriptome data to ensure that each transcript encompassed the entire open reading frame, allowing us to assess expression from approximately the same relative position on each transcript. A further BLASTn search against the *P. bursaria*–*Chlorella* spp. host and endosymbiont transcript datasets confirmed that each qPCR amplicon site was specific to the host. Using mRNA extracted from *P. bursaria* during *Dcr1*-RNAi feeding, and qPCR amplicons adjacent to the equivalent position of the Dcr1A/B dsRNA target site, qPCR revealed no significant knock-down in *Dcr2/3*, *Dcl1/2*, *Dcl3/4* or *Dcl5* transcripts in response to Dcr1A and Dcr1B dsRNA exposure (figure 3*b*). Indeed, we noted an increase in *Dcr2/3*, *Dcl1/2* and *Dcl3/4* expression suggesting that these transcripts are potentially being upregulated to compensate for reduced *Dcr1* expression. These data confirm that exposure to Dcr1A/Dcr1B dsRNA results in specific knock-down of host *Dcr1* in *P. bursaria*.

To understand the effect of *Dcr1* knock-down on endogenously triggered *P. bursaria* RNAi function, a size distribution of global host-derived sRNA abundance was compared between cultures exposed to Dcr1A and Dcr1B dsRNA, or a non-hit 'scramble' dsRNA control (figure 3*c*). A significant reduction in both 23-nt sense and antisense sRNA reads was observed upon Dcr1A/Dcr1B dsRNA exposure, accompanied by no significant reduction in any other sRNA read size between 20- and 30-nt. These data demonstrate that delivery of Dcr1A/Dcr1B dsRNA results in a specific reduction in endogenous 23-nt siRNA abundance, indicative of disruption of predicted *Dcr1* function. An increase in all greater

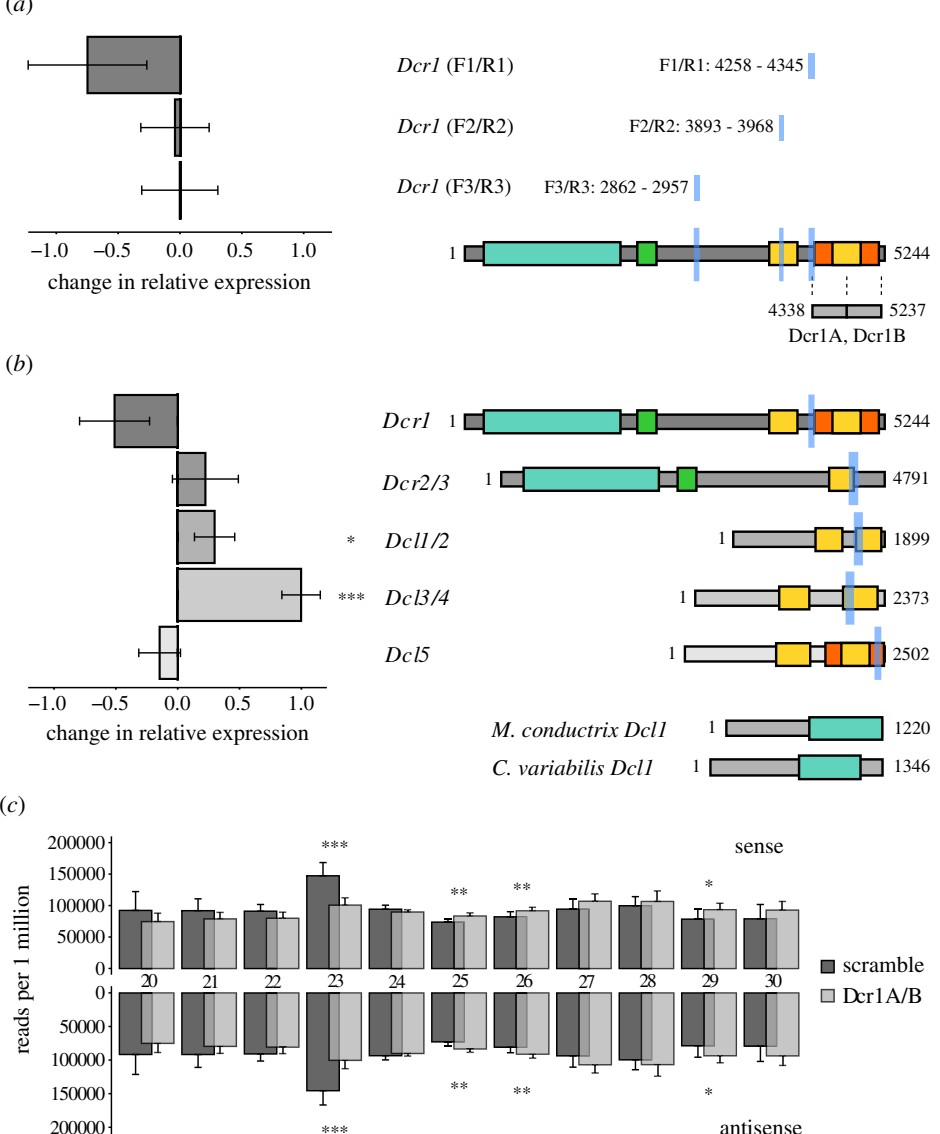

**Figure 3.** Investigating *Dcr1* function. (*a*) qPCR of mRNA extracted from day 3 of Dicer-RNAi feeding, revealing *Dcr1* gene knock-down in *P. bursaria* in response to Dcr1A and Dcr1B dsRNA exposure. A schematic of *P. bursaria Dcr1* shows the target sites of tandem 500-nt Dcr1A and Dcr1B dsRNA constructs, and amplicon sites of respective *Dcr1* F1/R1, F2/R2 and F3/R3 qPCR primers (light blue). Note the proximity of the amplicon site to the dsRNA target site, relative to the degree of knock-down detected via qPCR. (*b*) Additional qPCR of mRNA extracted from day 3 of Dicer-RNAi feeding. Knock-down was not observed in *Dcr2/3*, *Dd1/2* or *Dd3/4*, and is inconclusive in *Dd5*. A schematic of all *P. bursaria* Dicer or Dicer-like transcripts shows functional domain homology, and amplicon sites of respective qPCR primers (light blue) for each transcript. For all qPCR data, change in relative expression (ddCt) was calculated for treated (Dcr1A/Dcr1B dsRNA) versus untreated (scramble dsRNA) control cultures, and normalized against the standardized change in expression of a *GAPDH* housekeeping gene. Data are represented as mean ± s.e.m. of six unpaired biological replicates, per treatment. Unpaired dCt values were pooled and averaged prior to calculation of ddCt. Error bars for ddCt values were propagated from the s.e.m. of dCt values. Raw Ct, dCt and error propagation calculations for all ddCt values are available in electronic supplementary material, tables S4 and S5. Significance for qPCR calculated as $^*p \leq 0.05$, $^{***}p \leq 0.001$ using a paired *t*-test. For all schematic domains: turquoise, Helicase; green, Dicer dimer; yellow, RIBOc; orange, RNC. *Micractinium conductrix* Dcl1 and *Chlorella variabilis* Dcl1 demonstrates the divergence of Dicer homologues in the algal endosymbionts of *P. bursaria*. Amino acid alignment data for putative *P. bursaria* homologues used in the above datasets are available on Figshare (https://doi.org/10.6084/m9.figshare.13387631.v1). For phylogenetic analysis confirming the identity of these Dicer and Dicer-like components in *P. bursaria*, see electronic supplementary material, datasets S1 and S2. (*c*) Overlaid size distribution of sense and antisense sRNAs mapping to *P. bursaria* host transcripts during exposure to Dcr1A and Dcr1B dsRNA (light grey), or a non-hit 'scramble' dsRNA control (dark grey). sRNA was sequenced from *P. bursaria* after 7, 8 and 9 days of *E. coli* vector-based RNAi feeding to deliver respective dsRNA. Note the reduction in 23-nt sense and antisense reads upon Dcr1A/Dcr1B dsRNA exposure, accompanied by an increase in $\geq$25-nt sense and antisense reads. Data are presented as mean ± s.d. of nine biological replicates (three per time point), and normalized to reads per 1 million 20–30-nt host-mapping reads, per sample. Significance for sRNA abundance calculated as $^*p \leq 0.05$, $^{**}p \leq 0.01$ and $^{***}p \leq 0.001$ using a paired *t*-test.

than or equal to 25-nt sRNA reads upon Dcr1A/Dcr1B dsRNA exposure (figure 3c) may correspond to the increased expression of *Dcr2/3*, *Dcl1/2* and *Dcl3/4* transcripts observed in figure 3b (see also table 1), further corroborating that these additional Dicer or Dicer-like components are potentially being upregulated in *P. bursaria* to compensate for disruption of *Dcr1* function. Alternatively, an increase in 25-nt sRNA reads (resembling scnRNA), and *Dcl1/2* and *Dcl3/4* expression, may be due to increased autogamy in *P. bursaria*, suggesting that disruption of *Dcr1* function could result in host cellular stress [42,64]. Nonetheless, the absence of significant reduction in all other sRNA sizes (with the exception of 23-nt) indicates that knock-down in response to Dcr1A/Dcr1B dsRNA exposure is effective in specifically reducing host *Dcr1* function in *P. bursaria*.

## 2.4. Validation of *Dcr1*, *Piwi* and *Pds1* function

Finally, we sought to corroborate the function of a putative feeding-induced siRNA-based RNAi pathway in *P. bursaria*. In an attempt to disrupt *P. bursaria* siRNA-based RNAi function, we exposed cultures to Dcr1A/Dcr1B dsRNA during *u2af1*-RNAi feeding. Simultaneous knock-down of *Dcr1* during *u2af1*-RNAi feeding gave rise to an 'RNAi rescue' phenotype, restoring *P. bursaria* culture growth in *u2af1* dsRNA exposed cultures (figure 4a). Importantly, this effect was significantly greater than the same relative simultaneous delivery of an empty vector control during *u2af1*-RNAi feeding, indicating that rescue of *P. bursaria* culture growth was not due to dilution of *u2af1* dsRNA template.

We next aimed to corroborate the function of three feeding-induced siRNA-based components that showed low resolution in the phylogenetic trees in *P. bursaria*: *PiwiA1*, *PiwiC1* and *Pds1*. As for *Dcr1*, simultaneous knock-down of either *Pds1*, *PiwiA1* or *PiwiC1* during *u2af1*-RNAi feeding each gave rise to an 'RNAi recue' phenotype, restoring *P. bursaria* culture growth in *u2af1* dsRNA exposed cultures (figure 4b). *Pds1* is a *Paramecium*-specific component of feeding-induced siRNA-based RNAi first discovered in *P. tetraurelia* [33]. Sequence homology searches of known functional protein domains could not ascribe a putative function to Pds1; however, it was suggested that this protein may play a role in the export of dsRNA from the digestive-phagocytic vacuole into the host cytoplasm [33,36]. Our confirmation that feeding-induced siRNA-based RNAi in *P. bursaria* is dependent upon *Pds1* is important. As the sampled green algae do not encode an identifiable homologue of *Pds1*, this reiterates that the RNAi effect we have observed is derived from the *P. bursaria* host, and not from the algal endosymbiont.

Delivery of *Dcr1*, *PiwiA1*, *PiwiC1* or *Pds1* dsRNA to perturb siRNA-based RNAi function will never provide a complete 'RNAi rescue', since they are themselves important for cellular function. Indeed, partial knock-down of *Dcr1* in this manner may explain why reduction in *Dcr1* transcript expression was not deemed to be statistically significant via qPCR (figure 3b). Mutagenesis screens in *P. tetraurelia* have previously revealed that *Dcr1* null alleles typically result in lethality [33], suggesting that these pathway components have essential functions in *Paramecium*. We propose that partial knock-down of *Dcr1* via an *E. coli* feeding vector-based 'paradox' approach is therefore preferable to total silencing that would otherwise kill the cell. Perturbation of RNAi through disruption of *Dcr1* (knock-down, rather than knock-out) is sufficient to attenuate the RNAi effect, and thereby provide an appropriate control for inferring *bona fide* RNAi-mediated knock-down of an alternative primary gene target using a feeding-based approach. We have demonstrated that disruption of these essential RNAi components is effective at perturbing both background endogenous (*Dcr1*; figure 3c) and exogenously triggered (*Dcr1*, *PiwiA1*, *PiwiC1* and *Pds1*; figure 4a,b) siRNA pathways in *P. bursaria*. Taken together, these data confirm that feeding-induced siRNA-based RNAi in *P. bursaria* is dependent upon host *Dcr1*, *PiwiA1*, *PiwiC1* and *Pds1* function.

## 3. Discussion

Here, we have identified the repertoire of cognate RNAi components present in *P. bursaria*, including essential proteome constituents of the siRNA-, scnRNA- and iesRNA-based RNAi pathways. These include orthologues of the pathway components; Dicer, Dicer-like, Piwi, Rdr, Cid and Pds1 that are present in the non-photo-endosymbiotic model system, *P. tetraurelia*. Our comparison across the *Paramecium* clade (figure 1) reveals that many of these components probably originated from the WGD event that occurred prior to the radiation of the *Paramecium aurelia* species complex, which diverged separately from the *P. bursaria* lineage. Importantly, an unusually large number of copies of RNAi-component encoding genes have been retained in the *Paramecium* clade (greater than 80%

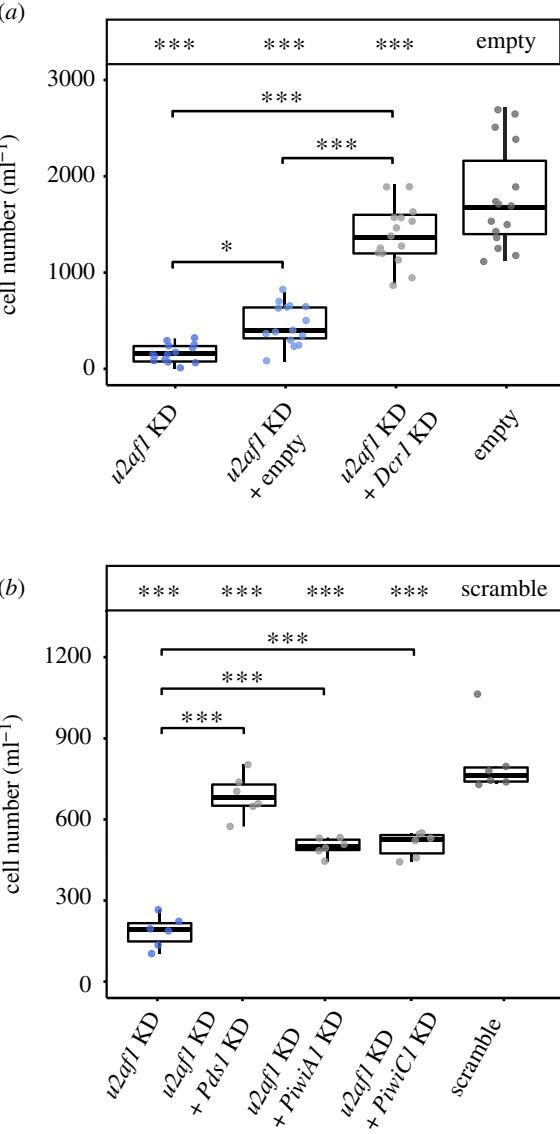

**Figure 4.** Validation of *Dcr1*, *Piwi* and *Pds1* function. (a) *Paramecium bursaria* cell number after 10 days of feeding with HT115 *E. coli* expressing either: *u2af1* dsRNA (dark blue); *u2af1* dsRNA mixed with empty vector (light blue) or *Dcr1* dsRNA (grey); or an empty vector control (dark grey). (b) *Paramecium bursaria* cell number after 12 days of feeding with HT115 *E. coli* expressing either: *u2af1* dsRNA (dark blue); *u2af1* dsRNA mixed with *Pds1*, *PiwiA1* or *PiwiC1* dsRNA (grey); or a 'scramble' control (dark grey). Multiple vector delivery was conducted at a 50 : 50 ratio during feeding. Significance calculated as $^*p \leq 0.05$, $^{**}p \leq 0.001$ using a generalized linear model with quasi-Poisson distribution. Boxplot data are represented as max, upper quartile (Q3), mean, lower quartile (Q1) and min values of five biological replicates. Asterisks in the box above each plot correspond to significance compared with empty vector, or 'scramble' dsRNA controls, respectively. Individual data points are shown. Confirmation of the consistent effect of empty vector compared with 'scramble' dsRNA exposure are shown in electronic supplementary material, figure S2.

retention for all components; table 2), exceeding the 40–60% retention rate observed in paralogues of this WGD event across the *Paramecium aurelia* species complex [40]. This observation suggests that these RNAi components are either highly expressed, and thus retention is enforced by gene dosage constraints, and/or have undergone significant neo- or sub-functionalization that has driven retention of these paralogues following the initial WGD event [65]. Our phylogenetic analysis of Dicer, RdRP and AGO-Piwi components (electronic supplementary material, datasets S1–S4) supports the occurrence of at least three WGD events within the ciliate group [38–40]. These are hypothesized to have occurred (i) after the divergence of the CONThreeP clade (Colpodea, Oligohymenophorea, Nassophorea, Prostomatea, Plagiopylea and Phyllopharyngea) [60,66] from *Oxytricha trifallax* (Spirotrichea) and the broader ciliates, (ii) after the divergence of *Paramecium* from *Tetrahymena*

**Table 2.** Calculation of RNAi component-encoding gene retention rate across the *Paramecium aurelia* species complex (compared across 11 available genome projects).

| *Paramecium bursaria* homologue | putative *P. tetraurelia* homologues[a] | predicted copies in *P. aurelia* species complex | actual copies in *P. aurelia* species complex[b] | copy loss in *P. aurelia* species complex | gene retention rate (%)[c] |
|---|---|---|---|---|---|
| *Dcr1* | 1 | 11 | 11 | 0 | 100.00 |
| *Dcr2/3* | 2 | 22 | 20 | −2 | 90.91 |
| *Dcl1/2* | 2 | 22 | 21 | −1 | 95.45 |
| *Dcl3/4* | 2 | 22 | 21 | −1 | 95.45 |
| *Dcl5* | 1 | 11 | 11 | 0 | 100.00 |
| *Pds1* | 1 | 11 | 11 | 0 | 100.00 |
| *Rdr1/4* | 2 | 22 | 21 | −1 | 95.45 |
| *Rdr2* | 1 | 11 | 11 | 0 | 100.00 |
| *Rdr3* | 1 | 11 | 11 | 0 | 100.00 |
| *PiwiA1* | 5 | 55 | 51 | −4 | 92.73 |
| *PiwiA2* | 3 | 33 | 31 | −2 | 93.94 |
| *PiwiB* | 1 | 11 | 9 | −2 | 81.82 |
| *PiwiC1*[d] | 3 | 33 | 32 | −1 | 96.97 |
| *PiwiC2* | 2 | 22 | 20 | −2 | 90.91 |
| *PiwiD* | 2 | 22 | 20 | −2 | 90.91 |
| *Cid1/3* | 2 | 22 | 20 | −2 | 90.91 |
| *Cid2* | 1 | 11 | 9 | −2 | 81.82 |

[a]For phylogenetic identification of *Paramecium* homologues, see electronic supplementary material, datasets S1–S6.
[b]Actual copy number taken from figure 1.
[c]Not including locally duplicated copies of *Dcl5*, *Rdr1*, *Rdr2* and *Piwi03*.
[d]*Piwi04c* excluded due to poor phylogenetic resolution.

*thermophilia* and the remaining Oligohymenophorea, and (iii) after the divergence of the *Paramecium aurelia* species complex from the remainder of the *Paramecium* clade (*Paramecium caudatum* and *Paramecium bursaria*).

Using an *E. coli* vector feeding-based approach for induction of RNAi, we have demonstrated that knock-down of a conserved splicing factor, *u2af1*, results in *P. bursaria* culture growth retardation. Segregation of host germline and somatic nuclei in *Paramecium* makes long-term stable (conventional) transformation methods inconsistent and therefore unfeasible for systematic functional genomic profiling. *Paramecium* species are known to conjugate through sexual reproduction approximately every 200 generations [67,68]. This means that a library of somatic transformants (featuring RNAi deficient mutations) would need to be maintained and propagated through mitosis to prevent these genetic changes from being lost upon regeneration of the somatic macronucleus [64]. We therefore propose that delivery of exogenously derived dsRNA complementary to a target transcript, in the manner conducted in this study and others [48–51], remains the optimal experimental approach for large-scale gene knock-down surveys in this, and possibly other, ciliate systems.

Finally, we have corroborated the function of several RNAi components; including *Dcr1*, two unduplicated AGO-Piwi factors (*PiwiA1* & *PiwiC1*) and *Pds1*, via simultaneous component knock-down to rescue *P. bursaria* culture growth, supporting the hypothesis that these factors are required for exogenously induced siRNA-based RNAi induction in this system. We have demonstrated that, though the algal endosymbiont encodes a putative RNAi pathway including a *Dcl1* homologue, these do not appear to generate sRNAs in the same size range as the host (electronic supplementary material, figure S1). This, together with our assessment of the *Paramecium*-specific factor, *Pds1*, further reinforces that any RNAi effect initiated through a feeding-based approach is host-derived. The data presented in this study have allowed us to de-convolute a functional exogenously inducible siRNA-based RNAi pathway in the endosymbiotic ciliate, *P. bursaria*. We hope that these results will further

promote the use of the *P. bursaria–Chlorella* spp. endosymbiosis as a key model system to investigate the genetic basis of a nascent endosymbiotic cell–cell interaction.

# 4. Methods

## 4.1. Culture conditions and media

In all RNAi experiments, *Paramecium bursaria* 186b (CCAP 1660/18) strain was used. For genome analysis, *P. bursaria* 186b strain was used. For transcriptome analysis, *P. bursaria* 186b, CCAP 1660/12 and Yad1g1N strains were used.

 *P. bursaria* cells were cultured in New Cereal Leaf–Prescott Liquid medium (NCL). NCL medium was prepared by adding 4.3 mg l$^{-1}$ CaCl$_2$.2H$_2$O, 1.6 mg l$^{-1}$ KCl, 5.1 mg l$^{-1}$ K$_2$HPO$_4$, 2.8 mg l$^{-1}$ MgSO$_4$.7H$_2$O to deionized water. Wheat bran (1 g l$^{-1}$) was added, and the solution boiled for 5 min. Once cooled, medium was filtered once through Whatman Grade 1 filter paper and then through Whatman GF/C glass microfibre filter paper. Filtered NCL medium was autoclaved at 121°C for 30 min to sterilize prior to use.

 NCL medium was bacterized with *Klebsiella pneumoniae* SMC and supplemented with 0.8 mg l$^{-1}$ β-sitosterol prior to propagation. *P. bursaria* cells were sub-cultured 1:9 into fresh bacterized NCL medium once per month, and maintained at 18°C with a light–dark (LD) cycle of 12:12 h.

## 4.2. Transcriptome analysis

RNA was extracted from *P. bursaria* 186b for transcriptome analysis, using approximately 10$^6$ host cells from five replicates at three time points over the 12:12 h LD cycle (6 h L, 1.5 h D and 10.5 h D). RNA extraction was performed using the RNA PowerSoil Total RNA Isolation Kit (MoBio) following the manufacturer's protocol. Samples were checked for quality using an Agilent TapeStation (High Sensitivity RNA ScreenTape) and a NanoDrop ND-1000, resulting in four low-quality samples which were discarded (RNA Integrity Number less than 1, NanoDrop concentration less than 15 ng µl$^{-1}$ or TapeStation less than 450 ng total). RNA for the remaining 11 samples (four 6 h L, four 1.5 h D and three 10.5 h D) was matched to 400 ng, and library preparation performed using the TruSeq Stranded Total RNA Kit (Illumina) following the manufacturer's protocol. Prepared libraries of 11 samples were then sequenced using a paired-end 120-bp rapid run across two lanes on an Illumina HiSeq 2500, yielding approximately 1112 million reads (with a mean of 101 million reads per sample [s.e.m. of 2.9 million reads]). For details of additional transcriptome sequence acquisition from *P. bursaria* CCAP 1660/12, which also included 'single-cell' transcriptome analyses, please refer to the electronic supplementary material, Methods.

 Raw reads were trimmed at Q5 in Trimmomatic (v. 0.32) [69]. Reads were then error corrected using rcorrector (v. 1.0.0) and digitally normalized using Khmer v. 1.4.1 [70] at a k-mer size of 20 and average coverage of 20. The remaining reads were then assembled using rnaSPAdes (v. 3.11.1) [71] and Trinity (v. 2.0.2) [72]. On the basis of RSEM (v. 1.2.24) [73] and assembly statistics, the Trinity assembly was selected for further analysis.

 ORFs were called from Trinity assembled transcripts using Transdecoder, using both ciliate (*Tetrahymena*) and universal encodings. The longest peptide sequences were retained for each. The remaining ORFs were then annotated via a BLASTX (v. 2.2.31) search against a genome database consisting of: *Arabidopsis thaliana*, *Aspergillus nidulans*, *Bacillus cereus* ATCC 14579, *Burkholderia pseudomallei* K96243, *Candidatus Korarchaeum cryptofilum* OPF8, *Chlamydomonas reinhardtii*, *Chlorella variabilis* NC64A, *Chlorella vulgaris* C-169, *Escherichia coli* str. K-12 substr. MG1655, *Homo sapiens*, *Methanococcus maripaludis* S2, *Oxytricha trifallax*, *Paramecium biaurelia*, *P. caudatum*, *P. multimicronucleatum*, *P. primaurelia*, *P. sexaurelia*, *P. tetraurelia*, *Saccharomyces cerevisiae* S288C, *Streptomyces coelicolor* A32, *Sulfolobus islandicus* M.14.25, *Tetrahymena borealis*, *T. elliotti*, *T. malaccensis*, *T. thermophila* macronucleus, *T. thermophila* micronucleus and *Ustilago maydis*.

 Assembled transcripts were subsequently binned into either 'host', 'endosymbiont', 'food' or 'other' datasets, using a phylogeny-based machine-learning approach (https://github.com/fmaguire/dendrogenous, see electronic supplementary material, Methods). Binned sequences were further annotated using SignalP (v. 4.0), TMHMM (v. 2.0) and BLAST2GO (v. 4). Each dataset was filtered to remove any sequences with a predicted peptide sequence shorter than 30 amino acids.

*P. bursaria* Yad1g1N [22] transcriptome reads were downloaded from DDBJ (Submission DRA000907), and processed using the same approach to assembly and binning as the *P. bursaria* 186b dataset described above.

## 4.3. Phylogenetic analysis

Assembled datasets of ciliate encoded predicted proteins ('host' bin) and universally encoded predicted proteins ('endosymbiont' bin) were searched using BLASTp and a minimum expectation of $1 \times 10^{-5}$, to identify homologues of annotated protein sequences that are putatively encoded by both host and endosymbiont. Proteins predicted from genomic data were downloaded from ParameciumDB [74] for *Paramecium biaurelia*, *P. caudatum*, *P. decaurelia*, *P. dodecaurelia*, *P. jenningsi*, *P. novaurelia*, *P. octaurelia*, *P. primaurelia*, *P. quadecaurelia*, *P. sexaurelia*, *P. tetraurelia* and *P. tredecaurelia*. These were added to a curated dataset of genomic and transcriptomic data from a further 41 ciliate species [75] to assess for homologues throughout the ciliates. Identified homologues were checked against the NCBI non-redundant protein sequences (nr) database via reciprocal BLASTp search.

Protein sequences were aligned using MAFFT [76] (v. 7.471) and masked using TrimAL [77] (v. 1.4.rev15) allowing for no gaps. Sequences were manually checked in SeaView [78] (v. 5.0.4), and highly divergent or identical sequences from the same genomic source were removed. Phylogenies were generated using IQ-TREE (v. 2.0.3) with 1000 non-parametric non-rapid bootstraps, using the best fit substitution model calculated with IQ-TREE's inbuilt ModelFinder implementation and according to the Bayesian inference criterion (BIC). The models chosen for tree generation are listed in the respective figure legends.

## 4.4. Gene synthesis and construct design

Sequences for plasmid constructs were synthesized *de novo* by either Genscript or SynBio Technologies, and cloned into an L4440 plasmid vector (Addgene plasmid #1654). Sequences and cloning sites for each plasmid construct are detailed in electronic supplementary material, table S1. All modified constructs were confirmed by Sanger sequencing (Eurofins Genomics).

## 4.5. RNAi feeding

*P. bursaria* was fed with *E. coli* transformed with an L4440 plasmid construct with paired IPTG-inducible T7 promoters, facilitating targeted gene knock-down through the delivery of complementary dsRNA. L4440 plasmid constructs were transformed into *E. coli* HT115 competent cells and grown overnight on LB agar (50 µgml$^{-1}$ Ampicillin and 12.5 µgml$^{-1}$ Tetracycline) at 37°C. Positive transformants were picked and grown overnight in LB (50 µgml$^{-1}$ Ampicillin and 12.5 µgml$^{-1}$ Tetracycline) at 37°C with shaking (180 r.p.m.). Overnight pre-cultures were back-diluted 1 : 25 into 50 ml of LB (50 µgml$^{-1}$ Ampicillin and 12.5 µgml$^{-1}$ Tetracycline) and incubated for a further 2 h under the same conditions, until an OD$_{600}$ of between 0.4 and 0.6 was reached. *E. coli* cultures were then supplemented with 0.4 mM IPTG to induce template expression within the L4440 plasmid, and incubated for a further 3 h under the same conditions. *E. coli* cells were pelleted by centrifugation (3100 *g* for 2 min), washed with sterile NCL medium, and pelleted once more. *E. coli* cells were then re-suspended in NCL medium supplemented with 0.4 mM IPTG, 100 µgml$^{-1}$ Ampicillin and 0.8 µgml$^{-1}$ β-sitosterol, and adjusted to a final OD$_{600}$ of 0.1.

*P. bursaria* cells were pelleted by gentle centrifugation in a 96-well plate (10 min at 800 *g*), taking care not to disturb the cell pellet by leaving 50 µl of supernatant, and re-suspended 1 : 4 into 200 µl of induced *E. coli* culture medium (to make 250 µl total). Feeding was conducted daily for up to 14 days using freshly prepared bacterized medium.

## 4.6. qPCR analysis

RNA was extracted from *P. bursaria* 186b for gene expression analysis after three days of RNAi feeding. *P. bursaria* cells (approx. $10^3$ per culture) were pelleted by gentle centrifugation (800 *g* for 10 min), snap-frozen in liquid nitrogen, and stored at −80°C. RNA extraction was performed using TRIzol reagent (Invitrogen), following the manufacturer's protocol after re-suspending each pellet in 900 µl TRIzol reagent. RNA was precipitated using GlycoBlue Co-precipitant (Invitrogen) to aid RNA pellet

visualization, and then cleared of residual DNA using the TURBO DNA-*free* Kit (Ambion), following the manufacturer's protocol for routine DNase treatment.

RNA was reverse transcribed into single-stranded cDNA using the SuperScript® III First-Strand Synthesis SuperMix (Invitrogen), following the manufacturer's protocol. Quantitative PCR (qPCR) was performed in a StepOnePlus Real-Time PCR system (Thermo Fisher Scientific). Reaction conditions were optimized using a gradient PCR, with a standard curve determined using 10-fold dilutions of *P. bursaria* cDNA: *u2af1* (slope: −3.525; $R^2$: 0.994; efficiency: 92.157%), *dcr1* (slope: −3.400; $R^2$: 0.998; efficiency: 96.862%), *dcr2/3* (slope: −3.395; $R^2$: 0.996; efficiency: 97.050%), *dcl1/2* (slope: −3.494; $R^2$: 0.999; efficiency: 93.281%), *dcl3/4* (slope: −3.280; $R^2$: 0.999; efficiency: 101.767%), *dcl5* (slope: −3.411; $R^2$: 0.999; efficiency: 96.416%) and *GAPDH* (slope: −3.427; $R^2$: 1.000; efficiency: 95.802%), using StepOne software v. 2.3. Each 20 µl reaction contained 10 µl PowerUp SYBR Green Master Mix (Thermo Fisher Scientific), 500 nM of each primer and 1 µl (50 ng) cDNA. Primers pairs for each reaction are listed in electronic supplementary material, table S2. Each reaction was performed in duplicate for each of three biological replicates, alongside a 'no-RT' (i.e. non-reverse transcribed RNA) control to detect any genomic DNA contamination. Cycling conditions were as follows: UDG activation, 2 min at 50°C and DNA polymerase activation, 2 min at 95°C, followed by 40 cycles of 15 secs, 95°C and 1 min at 55–65°C (*u2af1* (57°C), *dcr1* (60°C), *dcr2/3* (60°C), *dcl1/2* (60°C), *dcl3/4* (60°C), *dcl5* (60°C) and *GAPDH* (60°C)). Each reaction was followed by melt-curve analysis, with a 60–95°C temperature gradient ($0.3°C\ s^{-1}$), ensuring the presence of only a single amplicon, and ROX was used as a reference dye for calculation of $C_T$ values. $C_T$ values were then used to calculate the change in gene expression of the target gene in RNAi samples relative to control samples, using a derivation of the $2^{-\Delta\Delta CT}$ algorithm [79].

## 4.7. sRNA isolation and sequencing

Total RNA for sRNA sequencing was extracted from *P. bursaria* (or free-living algal) cultures using TRIzol reagent (Invitrogen), as detailed above. To isolate sRNA from total RNA, samples were size separated on a denaturing 15% TBE-urea polyacrylamide gel. Gels were prepared with a 15 ml mix with final concentrations of 15% acrylamide/Bis (19 : 1), 8 M urea, TBE (89 mM Tris, 89 mM borate, 2 mM EDTA), and the polymerization started by the addition of 150 µl 10% APS (Sigma-Aldrich) and 20 µl TEMED (Sigma-Aldrich). Gels were pre-equilibrated by running for 15 min (200 V, 30 mA) in TBE before RNA loading. The ladder mix consisted of 500 ng ssRNA ladder (50–1000-nt, NEB#N0364S), and 5–10 ng of each 21 and 26-nt RNA oligo loaded per lane. The marker and samples were mixed with 2X RNA loading dye (NEB) and heat denatured at 90°C for 3 min before snap cooling on ice for 2 min prior to loading. Blank lanes were left between samples/replicates to prevent cross-contamination during band excision. Gels were then run for 50 min (200 V, 30 mA).

Once run, gels were stained by shaking (60 r.p.m.) for 20 min at RT in a 40 ml TBE solution containing 4 µl SYBR® Gold Nucleic Acid Gel Stain. Bands of the desired size range (approx. 15–30-nt) were visualized under blue light, excised and placed into a 0.5 ml tube pierced at the bottom by a 21-gauge needle, resting within a 1.5 ml tube, and centrifuged (16 000 $g$ for 1 min). Four hundred microlitres of RNA elution buffer (1 M Sodium acetate pH 5.5 and 1 mM EDTA) was added to the 1.5 ml tube containing centrifuged gel slurry, and the empty 0.5 ml tube discarded. Gel slurry was manually homogenized until dissolved using a 1 ml sterile plunger and incubated at RT for 2 h with shaking at 1400 r.p.m.

Solutions containing RNA elution buffer and gel slurry were transferred to a Costar Spin-X 0.22 µm filter column and centrifuged (16 000 $g$ for 1 min). The filter insert containing acrylamide was discarded. One millilitre of 100% EtOH was added to each solution, alongside 15 µg of GlycoBlue™ Coprecipitant (Invitrogen) to aid sRNA pellet visualization, and stored overnight at −80°C to precipitate. Precipitated solutions were centrifuged at 4°C (12 000 $g$ for 30 min), and the supernatant discarded. sRNA pellets were washed with 500 µl of cold 70% EtOH (12 000 $g$ for 15 min at 4°C), and air dried in a sterile PCR hood for 10 min, before re-suspending in 15 µl of RNAse-free water and storage at −80°C.

## 4.8. sRNA-seq and read processing

sRNA concentrations were determined using an Agilent 2100 Bioanalyzer, following the Agilent Small RNA kit protocol, and all samples matched to 0.7 ngml$^{-1}$ prior to sequencing. Library preparation and subsequent RNA-seq was performed for 54 samples using 50-bp paired-end, rapid run across four lanes on an Illumina HiSeq 2500, yielding approximately 120–150 million paired-end reads per lane (approx. 9–11 million paired-end reads per sample).

The raw paired-end reads from the RNA-seq libraries were trimmed using Trim Galore in order to remove barcodes (4-nt from each 3′- and 5′-end) and sRNA adaptors, with additional settings of a phred-score quality threshold of 20 and minimum length of 16-nt. Results were subsequently checked with FastQC.

Trimmed reads were mapped against the 'host' or 'endosymbiont' dataset of assembled transcripts using the HISAT2 alignment program with default settings. Post-mapping, the BAM files were processed using SAMTOOLS and a set of custom scripts (https://github.com/guyleonard/paramecium) to produce a table of mapped read accessions and their respective read lengths. Size distributions of sRNA abundance for each sample were plotted using the R programming language packages; tidyverse, grid.extra and ggplot2 in R Studio (v. 1.3.1073).

Data accessibility. The raw reads generated during transcriptome and sRNA sequencing are available on the NCBI Sequence Read Archive (accessions: SAMN14932981, SAMN14932982). All other datasets are available on Figshare (https://doi.org/10.6084/m9.figshare.c.5241983.v1), under the relevant headings. Custom scripts for host and endosymbiont transcript binning [80] (https://github.com/fmaguire/dendrogenous, https://doi.org/10.5281/zenodo.4639294) and sRNA read processing [81] (https://github.com/guyleonard/paramecium, https://doi.org/10.5281/zenodo.4638888) are available on GitHub and archived within the Zenodo repository.

Authors' contributions. B.H.J., D.S.M. and T.A.R. conceived and designed the experiments. F.M. conducted transcriptome assembly and binning. B.H.J., D.S.M. and T.A.R. wrote the manuscript. B.H.J., D.S.M. and G.L. conducted experimental work and analysed the data. B.E.H., S.W. and J.D.E aided in conceptual and experimental design, and in conducting experimental work.

Competing interests. The authors declare no competing interests.

Funding. This work was primarily supported by an EMBO YIP award and a Royal Society University Research Fellowship (UF130382) and latterly by an ERC Consolidator Grant (CELL-in-CELL) to T.A.R. S.W. and J.D.E. were supported by awards from the Wellcome Trust (WT107791/Z/15/Z) and the Lister institute.

Acknowledgements. We thank Karen Moore and the University of Exeter Sequence Service for support with the various sequencing projects. We thank Éric Meyer, Institut de Biologie de l'Ecole Normale Supérieure, Paris, for advice during set-up of the *P. bursaria* RNAi approach.

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
