## [Peer Review File · Royal Society Open Science]

Review History

RSOS-210140.R0 (Original submission)

Review form: Reviewer 1

Is the manuscript scientifically sound in its present form?

Yes

Are the interpretations and conclusions justified by the results?

Yes

Is the language acceptable?

Yes

Do you have any ethical concerns with this paper?

No

Have you any concerns about statistical analyses in this paper?

Yes

Recommendation?

Accept with minor revision (please list in comments)

Comments to the Author(s)

Jenkins et al. report about the genetic repertoire of RNAi components in *Paramecium bursaria*. This single celled organism is especially interesting to understand the evolution of symbiosis due to the ability to host intracellular chlorella algae. This work here describes on the one hand the evaluation of the ability to use the technique of RNAi by feeding by both, screening the genome for putative candidate genes and the direct application of dsRNA producing bacteria, and on the other hand an evaluation of sRNAs by NGS with a deeper analysis of recursive RNAi by double feeding with dsRNA of RNAi components themselves. By this approach the authors conclude that the siRNAs appear to derive from the host and not from the symbiont.

The general work is extremely interesting and represents the basis for a deeper analysis of the symbiosis, e.g. for a further analysis of RNA transfer between symbiont and host. In addition it's more than worth for the scientific community that RNAi by feeding works in *P.bursaria* and especially the knowledge of the presence of individual RNAi components in different *Paramecium* species is an extremely important analysis and I appreciate the time consuming work invested.

In fact, I cannot find many negative criticisms in this paper. All analysis appear more than solid, and the RNAi experiments including the subsequent analysis of phenotype and small RNAs involves the necessary controls and represents solid data. In addition the paper is well written and easy to follow the logic.

I have only minor comments:

- Line 138 dependent upon splicing for macronuclear generation and transcription: its not clear what macronuclear generation means in context of splicing
- Line 154: E.coli vector uptake or better "dsRNA uptake" or "E.coli uptake".
- Figure 1: "Stacked genes (single or unduplicated orthologues) represent putative local gene 312 duplication" this is not that clear to me: Is there one or two genes in *P.bursaria* with homology e.g. to Ptiwi08 and 14 of *P.tetraurelia*: I would assume only one, as 08 and 14 result from the last WGD in the aurelia complex and there will unlikely be two in *bursaria*: maybe the visualization is a bit confusing.

- Figure 3: sRNA seq: Does *P.bursaria* undergo autogamy and could the increased number of 25nt reads due to scnRNAs?

The authors here normalized to total reads number. Maybe they could also try to normalize to total *bursaria* mapping read number to account for varying read numbers mapping to (i) chlorella and (ii) food bacteria. In the worst, but most interesting situation, Dicer silencing could affect a potential RNA interchange of host and symbiont and thus symbiont population alters, thus altering the mapping statistics. I would indeed also be interesting in a mapping statistic: what's the general percentage of reads mapping to the symbiont, the host, the food, the dsRNA, rRNA etc.

Review form: Reviewer 2**Is the manuscript scientifically sound in its present form?**

Yes

Are the interpretations and conclusions justified by the results?

Yes

Is the language acceptable?

Yes

Do you have any ethical concerns with this paper?

No

Have you any concerns about statistical analyses in this paper?

No

Recommendation?

Accept with minor revision (please list in comments)

Comments to the Author(s)

This manuscript represents a very interesting study. The authors investigated the repertoire of RNAi pathway protein-encoding genes in the model nascent endosymbiotic system, *Paramecium bursaria*-*Chlorella* spp. They identified essential proteome components of the small interfering RNA, scan RNA, and internal eliminated sequence RNA pathways by using comparative genomic and transcriptomic approaches that were supported by phylogenetic analyses. Their results support the presence of a functional, host derived, exogenously-induced siRNA-based RNAi pathway in the *P. bursaria*-*Chlorella* spp. endosymbiotic system, dependent on Dcr1, Piwi and Pds1 protein function. The study seems to be well conducted and the manuscript is well written. The following are my minor comments or suggestions.

- 1) In Figure 2C, It is not clear if there are any statistical differences in the *P. bursaria* cell number between the cultures fed with HT115 *E. coli* expressing u2af1 dsRNA (blue) and containing an empty vector control (grey). Although the figure legend indicates the difference by using three asterisks, these are not presented in the figure. In addition, I don't know if the use of blue and grey color is the best strategy in the figure. When the readers print out a copy of this paper with a black/white printer, it might be hard to tell the difference between these two colors.
- 2) In Figure 3C legend, the authors indicate "one-way analysis of variance (ANOVA)". Please make sure if this description is sufficient. My understanding is that they would need to use a different test (e.g., T-test) to tell the difference.

Decision letter (RSOS-210140.R0)

Dear Dr Jenkins

On behalf of the Editors, we are pleased to inform you that your Manuscript RSOS-210140 "Characterisation of the RNA-interference pathway as a Tool for Genetics in the Nascent Phototrophic Endosymbiosis, *Paramecium bursaria*" has been accepted for publication in Royal Society Open Science subject to minor revision in accordance with the referees' reports. Please find the referees' comments along with any feedback from the Editors below my signature.

We invite you to respond to the comments and revise your manuscript. Below the referees' and Editors' comments (where applicable) we provide additional requirements. Final acceptance of

your manuscript is dependent on these requirements being met. We provide guidance below to help you prepare your revision.

Please submit your revised manuscript and required files (see below) no later than 7 days from today's (ie 04-Mar-2021) date. Note: the ScholarOne system will 'lock' if submission of the revision is attempted 7 or more days after the deadline. If you do not think you will be able to meet this deadline please contact the editorial office immediately.

on behalf of Dr Steven Burgess (Associate Editor) and Malcolm White (Subject Editor)
openscience@royalsociety.org

Associate Editor Comments to Author (Dr Steven Burgess):
Comments to the Author:
Dear Dr Jenkins

Thank you for your submission, your manuscript has now been reviewed by two independent referees.

Both were highly supportive and found the study to be of interest and performed to a high standard.

There are minor suggestions relating to the statistical approach used requested.

Best
Steven

Reviewer comments to Author:
Reviewer: 1
Comments to the Author(s)

Jenkins et al. report about the genetic repertoire of RNAi components in *Paramecium bursaria*. This single celled organism is especially interesting to understand the evolution of symbiosis due to the ability to host intracellular chlorella algae. This work here describes on the one hand the evaluation of the ability to use the technique of RNAi by feeding by both, screening the genome for putative candidate genes and the direct application of dsRNA producing bacteria, and on the other hand an evaluation of sRNAs by NGS with a deeper analysis of recursive RNAi by double

feeding with dsRNA of RNAi components themselves. By this approach the authors conclude that the siRNAs appear to derive from the host and not from the symbiont.

The general work is extremely interesting and represents the basis for a deeper analysis of the symbiosis, e.g. for a further analysis of RNA transfer between symbiont and host. In addition it's more than worth for the scientific community that RNAi by feeding works in *P.bursaria* and especially the knowledge of the presence of individual RNAi components in different *Paramecium* species is an extremely important analysis and I appreciate the time consuming work invested.

In fact, I cannot find many negative criticisms in this paper. All analysis appear more than solid, and the RNAi experiments including the subsequent analysis of phenotype and small RNAs involves the necessary controls and represents solid data. In addition the paper is well written and easy to follow the logic.

I have only minor comments:

- Line 138 dependent upon splicing for macronuclear generation and transcription: its not clear what macronuclear generation means in context of splicing
- Line 154: E.coli vector uptake or better "dsRNA uptake" or "E.coli uptake".
- Figure 1: "Stacked genes (single or unduplicated orthologues) represent putative local gene 312 duplication" this is not that clear to me: Is there one or two genes in *P.bursaria* with homology e.g. to Ptiwi08 and 14 of *P.tetraurelia*: I would assume only one, as 08 and 14 result from the last WGD in the aurelia complex and there will unlikely be two in *bursaria*: maybe the visualization is a bit confusing..
- Figure 3: sRNA seq: Does *P.bursaria* undergo autogamy and could the increased number of 25nt reads due to scnRNAs?

The authors here normalized to total reads number. Maybe they could also try to normalize to total *bursaria* mapping read number to account for varying read numbers mapping to (i) *Chlorella* and (ii) food bacteria. In the worst, but most interesting situation, Dicer silencing could affect a potential RNA interchange of host and symbiont and thus symbiont population alters, thus altering the mapping statistics. I would indeed also be interesting in a mapping statistic: what's the general percentage of reads mapping to the symbiont, the host, the food, the dsRNA, rRNA etc.

Reviewer: 2

Comments to the Author(s)

This manuscript represents a very interesting study. The authors investigated the repertoire of RNAi pathway protein-encoding genes in the model nascent endosymbiotic system, *Paramecium bursaria*-*Chlorella* spp. They identified essential proteome components of the small interfering RNA, scan RNA, and internal eliminated sequence RNA pathways by using comparative genomic and transcriptomic approaches that were supported by phylogenetic analyses. Their results support the presence of a functional, host derived, exogenously-induced siRNA-based RNAi pathway in the *P. bursaria*-*Chlorella* spp. endosymbiotic system, dependent on Dcr1, Piwi and Pds1 protein function. The study seems to be well conducted and the manuscript is well written. The following are my minor comments or suggestions.

1) In Figure 2C, It is not clear if there are any statistical differences in the *P. bursaria* cell number between the cultures fed with HT115 *E. coli* expressing u2af1 dsRNA (blue) and containing an empty vector control (grey). Although the figure legend indicates the difference by using three asterisks, these are not presented in the figure. In addition, I don't know if the use of blue and grey color is the best strategy in the figure. When the readers print out a copy of this paper with a black/white printer, it might be hard to tell the difference between these two colors.

2) In Figure 3C legend, the authors indicate "one-way analysis of variance (ANOVA)". Please make sure if this description is sufficient. My understanding is that they would need to use a different test (e.g., T-test) to tell the difference.

===PREPARING YOUR MANUSCRIPT===

===PREPARING YOUR REVISION IN SCHOLARONE===

- 1) One version identifying all the changes that have been made (for instance, in coloured highlight, in bold text, or tracked changes);
 - 2) A 'clean' version of the new manuscript that incorporates the changes made, but does not highlight them.
 - An individual file of each figure (EPS or print-quality PDF preferred [either format should be produced directly from original creation package], or original software format).
 - An editable file of each table (.doc, .docx, .xls, .xlsx, or .csv).
 - An editable file of all figure and table captions.
- Note: you may upload the figure, table, and caption files in a single Zip folder.
- Any electronic supplementary material (ESM).
 - If you are requesting a discretionary waiver for the article processing charge, the waiver form must be included at this step.
 - If you are providing image files for potential cover images, please upload these at this step, and inform the editorial office you have done so. You must hold the copyright to any image provided.
 - A copy of your point-by-point response to referees and Editors. This will expedite the preparation of your proof.

- Ensure that your data access statement meets the requirements at <https://royalsociety.org/journals/authors/author-guidelines/#data>. You should ensure that you cite the dataset in your reference list. If you have deposited data etc in the Dryad repository, please only include the 'For publication' link at this stage. You should remove the 'For review' link.
- If you are requesting an article processing charge waiver, you must select the relevant waiver option (if requesting a discretionary waiver, the form should have been uploaded at Step 3 'File upload' above).
- If you have uploaded ESM files, please ensure you follow the guidance at <https://royalsociety.org/journals/authors/author-guidelines/#supplementary-material> to include a suitable title and informative caption. An example of appropriate titling and captioning may be found at https://figshare.com/articles/Table_S2_from_Is_there_a_trade-off_between_peak_performance_and_performance_breadth_across_temperatures_for_aerobic_scope_in_teleost_fishes_/3843624.

Author's Response to Decision Letter for (RSOS-210140.R0)

See Appendix A.

Decision letter (RSOS-210140.R1)

Dear Dr Jenkins,

It is a pleasure to accept your manuscript entitled "Characterisation of the RNA-interference pathway as a Tool for Reverse Genetic analysis in the Nascent Phototrophic Endosymbiosis, *Paramecium bursaria*" in its current form for publication in Royal Society Open Science.

on behalf of Dr Steven Burgess (Associate Editor) and Malcolm White (Subject Editor)
openscience@royalsociety.org

Appendix A

Dear Editors and Reviewers

Thank you for the careful consideration of our manuscript, we are delighted with the positive appraisal and constructive responses provided. Please find attached a point-by-point response to all comments. Our responses are written in black font. The review provided is written in blue font.

Thank you for your submission, your manuscript has now been reviewed by two independent referees.

Both were highly supportive and found the study to be of interest and performed to a high standard.

There are minor suggestions relating to the statistical approach used requested.

Thank you for the positive appraisal of our work. We have addressed all reviewer comments below.

Reviewer comments to Author:

Reviewer: 1

Comments to the Author(s)

Jenkins et al. report about the genetic repertoire of RNAi components in *Paramecium bursaria*. This single celled organism is especially interesting to understand the evolution of symbiosis due to the ability to host intracellular chlorella algae. This work here describes on the one hand the evaluation of the ability to use the technique of RNAi by feeding by both, screening the genome for putative candidate genes and the direct application of dsRNA producing bacteria, and on the other hand an evaluation of sRNAs by NGS with a deeper analysis of recursive RNAi by double feeding with dsRNA of RNAi components themselves. By this approach the authors conclude that the siRNAs appear to derive from the host and not from the symbiont.

The general work is extremely interesting and represents the basis for a deeper analysis of the symbiosis, e.g. for a further analysis of RNA transfer between symbiont and host. In addition it's more than worth for the scientific community that RNAi by feeding works in *P.bursaria* and especially the knowledge of the presence of individual RNAi components in different *Paramecium* species is an extremely important analysis and I appreciate the time consuming work invested. In fact, I cannot find many negative criticisms in this paper. All analysis appear more than solid, and the RNAi experiments including the subsequent analysis of phenotype and small RNAs involves the necessary controls and represents solid data. In addition the paper is well written and easy to follow the logic.

We thank the Reviewer for their positive comments. These RNAi component surveys were only possible courtesy of the extensive sequence resources already available for these *Paramecium* species. It was our pleasure to add to this growing knowledge base.

I have only minor comments:

- Line 138 dependent upon splicing for macronuclear generation and transcription: its not clear what macronuclear generation means in context of splicing

We have removed the ambiguous reference to macronuclear generation, to emphasise the role of splicing in pre-transcriptional intron removal. We have added a further reference here pertaining to intron-exon structure in ciliates.

- Line 154: E.coli vector uptake or better “dsRNA uptake” or “E.coli uptake”

Agreed, we have replaced this with “E. coli uptake”

- Figure 1: “Stacked genes (single or unduplicated orthologues) represent putative local gene 312 duplication” this is not that clear to me: Is there one or two genes in *P.bursaria* with homology e.g. to Ptiwi08 and 14 of *P.tetraurelia*: I would assume only one, as 08 and 14 result from the last WGD in the aurelia complex and there will unlikely be two in bursaria: maybe the visualization is a bit confusing..

I can appreciate the confusion here. Here, a horizontally ‘merged’ gene indicates a single unduplicated homologue related to the corresponding duplicated paralogs in *P. tetraurelia*. For example, the single Piwi08/14 orthologue in *P. bursaria*, which was later duplicated to give rise to Piwi08 and Piwi14 in *P. tetraurelia*. We have updated the figure legend to clarify this.

- Figure 3: sRNA seq: Does *P.bursaria* undergo autogamy and could the increased number of 25nt reads due to scnRNAs?

This was something that we had also wondered... Relating to the increase upon Dcr1 KD, we showed in Figure 3b that the activity of Dcl1/2 and Dcl3/4 (particularly Dcl3/4) are up-regulated in response to Dcr1 KD. As scnRNAs are produced by Dcl2 and Dcl3 in *P. tetraurelia*, we speculate that this could have resulted in the increased presence in the sRNA plot in Figure 3b. We have updated the text to reflect this consideration. This would certainly make for an interesting avenue of future investigation.

The authors here normalized to total reads number. Maybe they could also try to normalize to total bursaria mapping read number to account for varying read numbers mapping to (i) chlorella and (ii) food bacteria. In the worst, but most interesting situation, Dicer silencing could affect a potential RNA interchange of host and symbiont and thus symbiont population alters, thus altering the mapping statistics. I would indeed also be interesting in a mapping statistic: what’s the general percentage of reads mapping to the symbiont, the host, the food, the dsRNA, rRNA etc.

These reads were in fact normalised to total 20-30 nt host-mapping reads per sample to account for this, but we have updated the figure legend to emphasise this. Apologies for not being clearer here. In relation to the total reads per sample, we found that the number of reads mapping to the host were significantly greater than all other reads (bacterial/algal). For the host, a large proportion of

these belonged to host rRNA, followed by host mRNA. The second point about RNA interchange between symbiont and host is really very interesting, and is actually something we are currently looking into... we hope to release these data in a separate publication very soon.

Reviewer: 2

Comments to the Author(s)

This manuscript represents a very interesting study. The authors investigated the repertoire of RNAi pathway protein-encoding genes in the model nascent endosymbiotic system, *Paramecium bursaria*–*Chlorella* spp. They identified essential proteome components of the small interfering RNA, scan RNA, and internal eliminated sequence RNA pathways by using comparative genomic and transcriptomic approaches that were supported by phylogenetic analyses. Their results support the presence of a functional, host derived, exogenously-induced siRNA-based RNAi pathway in the *P. bursaria*-*Chlorella* spp. endosymbiotic system, dependent on Dcr1, Piwi and Pds1 protein function. The study seems to be well conducted and the manuscript is well written. The following are my minor comments or suggestions.

We thank the Reviewer for their positive comments.

1) In Figure 2C, It is not clear if there are any statistical differences in the *P. bursaria* cell number between the cultures fed with HT115 *E. coli* expressing *u2af1* dsRNA (blue) and containing an empty vector control (grey). Although the figure legend indicates the difference by using three asterisks, these are not presented in the figure. In addition, I don't know if the use of blue and grey color is the best strategy in the figure. When the readers print out a copy of this paper with a black/white printer, it might be hard to tell the difference between these two colors.

I agree with the point about the colour distinction, and we have updated Figure 2c with grey dashed lines to reflect this. We have additionally added asterisks to Figure 2c to indicate that the difference between *u2af1* dsRNA and empty vector fed *P. bursaria* cultures was significant at Day 12.

2) In Figure 3C legend, the authors indicate "one-way analysis of variance (ANOVA)". Please make sure if this description is sufficient. My understanding is that they would need to use a different test (e.g., T-test) to tell the difference.

The Reviewer is quite correct here, these have been updated with significance values calculated using a paired t-test.